A combination of hepatic leukemia factor and circulating tumor cells serve as effective biomarkers for lung adenocarcinoma prognosis

Zhi Yaofeng 1
Wu Jinhua 2
Li Ronggang 3
Chang Xuefei 1 4
Liu Silin 1 4
Lu Wenjie 1
Zheng Mingzhu 1
Liu Baoyi 1
Chen Jiarong 1 5
Zhang Xin zhangx45@mail3.sysu.edu.cn 1 6 7
Huang Yanming huangyanming_jxy@163.com 1 4
1 Clinical Experimental Center, Jiangmen Engineering Technology Research Center of Clinical Biobank and Translational Research, Jiangmen Key Laboratory of Precision and Clinical Translation Medicine, Jiangmen Central Hospital , Jiangmen , Guangdong , China
2 Department of Clinical Laboratory, Jiangmen Central Hospital , Jiangmen , Guangdong , China
3 Department of Pathology, Jiangmen Central Hospital , Jiangmen , Guangdong , China
4 Department of Pulmonary and Critical Care Medicine, Jiangmen Central Hospital , Jiangmen , Guangdong , China
5 Department of Oncology, Jiangmen Central Hospital , Jiangmen , Guangdong , China
6 Dongguan Key Laboratory of Medical Bioactive Molecular Developmental and Translational Research, Guangdong Provincial Key Laboratory of Medical Molecular Diagnostics, Guangdong Medical University , Dongguan , Guangdong , China
7 Collaborative Innovation Center for Antitumor Active Substance Research and Development, Guangdong Medical University , Dongguan , Guangdong , China
Ozdag Sevgili Hilal
Electronic publication date: 2025 Mar 18
Publication date: 2025
Volume: 13
Electronic Location ID: e19092
Received 2024 Jun 12; Accepted 2025 Feb 11
Copyright: ©2025 Zhi et al.
Copyright year: 2025
Copyright holder: Zhi et al.
License: This is an open access article distributed under the terms of the Creative Commons Attribution License, which permits unrestricted use, distribution, reproduction and adaptation in any medium and for any purpose provided that it is properly attributed. For attribution, the original author(s), title, publication source (PeerJ) and either DOI or URL of the article must be cited.
License URL: https://creativecommons.org/licenses/by/4.0/

Keywords: Lung adenocarcinoma, Biomarker, Prognosis, Hepatic leukemia factor, Circulating tumor cells

Funding: GuangDong Basic and Applied Basic Research Foundation 2021A1515220180 National Natural Science Foundation of China 82273384 82203748 This work was supported by the GuangDong Basic and Applied Basic Research Foundation (No. 2021A1515220180) and National Natural Science Foundation of China (No. 82273384 and No. 82203748). The funders had no role in study design, data collection and analysis, decision to publish, or preparation of the manuscript.

==============================
Background

Lung adenocarcinoma (LUAD) is a highly malignant tumor with the highest mortality rate among all cancers. Early diagnosis and prognosis are important factors in treatment. Hepatic leukemia factor (HLF) is thought to be closely associated with lung cancer metastasis. It is downregulated in lung cancer tissues and negatively correlated with the number of metastasis-activating circulating tumor cells (CTCs) in the peripheral blood of patients.

Method and Results

In this study, we analyzed data from LUAD samples in TCGA and found that HLF was significantly upregulated in samples with EGFR mutations. Immunohistochemical (IHC) staining of 343 clinical samples also revealed a trend of HLF upregulation in patients with EGFR mutations. EGFR is one of the driver genes in non-small cell lung cancer (NSCLC), and the proportion in LUAD is as high as 50% in the East Asian population. In this study, EGFR mutation was not significantly correlated with the prognosis of LUAD patients and the number of CTC was also not related to EGFR mutation, but was closely related to HLF expression, with more CTCs being captured in the peripheral blood of patients with low expression of HLF (SI ≤ 4). By following up these 343 LUAD patients, high HLF expression (SI > 4) was found to be an independent protective factor for progression-free survival regardless of EGFR status (P < 0.001), whereas high CTC count (> 3) was an independent risk factor for recurrence or death in LUAD patients (P < 0.001). When low HLF and high CTCs coexisted, patients had the shortest median survival time. Patients with low HLF or high CTCs appeared alone had a moderate median survival time. Patients had the longest median survival time when HLF was high and CTCs were low.

Conclusion

In summary, we believe that HLF expression in cancer tissues and the number of CTCs can be used as effective biomarkers for predicting the prognosis of LUAD, which plays an important role in clinical diagnosis and prognosis judgment.

Introduction

Lung cancer is a malignant tumor with the highest mortality rate (18%). Globally, it is the leading cause of death among male and female cancer patients (Sung et al., 2021). In China, respiratory neoplasms had the highest mortality rate from 2005 to 2020 with no sex differences in rates (Qi et al., 2023). Lung adenocarcinoma (LUAD) accounts for approximately 40% of all lung cancers and has a low overall survival rate (Wei et al., 2023). Therefore, early diagnosis, treatment, and prognosis are important factors. The circulating tumor DNA (ctDNA) assay is a part of liquid biopsy. ctDNA is a DNA fragment shed into the plasma from cancer cells, and its expression level partly predicts tumor burden and malignancy (Semenkovich et al., 2023; Bettegowda et al., 2014). ctDNA testing is widely used for lung cancer screening and prognostic monitoring.

Metastasis results in a mortality rate of up to 90% in lung cancer patients (Chaffer & Weinberg, 2011). According to the “seed and soil” theory proposed by Paget (1989), metastasis must be formed by actively growing tumor cells as the “seed”, overcoming apoptosis, and entering the peripheral bloodstream. Circulating tumor cells (CTCs) circulate in the peripheral blood and colonize local tissues to form metastatic cancer foci. Hepatic leukemia factor (HLF) is thought to be closely associated with tumor metastasis, and our previous study demonstrated that HLF was down-regulated in lung cancer tissues and negatively correlated with the number of metastatically active CTCs in the peripheral blood of patients. Downexpression of HLF promoted anaerobic metabolism to support anchorage-independent growth of NSCLC cells under low nutritional conditions by activating NF-κB/p65 signaling via disrupting translocation of PPARα and PPARβ (Chen et al., 2020). In a bioanalysis study (Ahmadi et al., 2024), HLF was found to be downregulated in pan-cancer malignant tissues and was related to apoptosis, cell cycle, EMT, and immunological infiltration. However, this study relied only on in-silico analyses and lacked in vivo and in vitro experiments to elucidate specific mechanisms. Therefore, we hypothesized that expression of HLF and the number of CTCs are important indicators for cancer prognosis. EGFR-mutation is one of the major drivers of LUAD and is an important target for current immunotherapies (Dantoing et al., 2021). With the development and application of tyrosine kinase inhibitors, the prognosis of lung tumors driven by specific mutations has been improving (Tan & Tan, 2022). It would be worthwhile to investigate whether the gene EGFR-mutation is directly associated with patient’s prognosis. ctDNA is easily detected in patients with gene mutations, especially with good specificity in EGFR-mutated NSCLC patients, and has been mainly used for molecular analysis to guide clinical drug use and the detection of drug resistance (Wang et al., 2021; Desai et al., 2024). However, the question remains: how can tumor prognosis be predicted more accurately in cancers without EGFR mutations or other mutation-driven genes? This study was devoted to exploring the effects of HLF, CTCs and EGFR mutations on the prognosis of LUAD patients, explaining their relationship with prognosis and guiding clinical diagnosis and treatment.

Materials & Methods

High-throughput data acquisition and analysis

The RNA sequencing and copy number variation profiles of LUAD from The Cancer Genome Atlas (TCGA; https://tcga-data.nci.nih.gov/tcga/) were downloaded. LUAD samples were divided into two groups of EGFRw.t. and EGFR mutations, for which the expression value of HLF was procured (the unit was RNA-Seq by Expectation Maximization, RSEM), and the log2 value of each sample was analyzed using Excel 2019.

Patients and clinical samples

A total of 343 cases of LUAD from 2017 to 2019 were collected from Jiangmen Central Hospital (Guangdong, China). Detailed follow-up information is provided in Table S1. In the follow-up data, tumor stage, grade, overall survival and progression-free survival were the key information. The tumor-node-metastasis (TNM) system (Lababede & Meziane, 2018; Telloni, 2017), includes: tumor size and local growth (T), extent of lymph node metastases (N), and occurrence of distant metastases (M), was used to define the extent of disease. In the analysis of tumor metastasis, NX, N0 and M0 were included as the group without metastasis, while N1, 2, 3 and M1 were included as the group with metastasis. The EGFR mutations were detected using HybriBio (Guangzhou, China; https://www.hybribio.cn/) using Sanger sequencing, and the mutation sites were compared to Cosmic (http://www.sanger.ac.uk/genetics/CGP/cosmic/). Full details are provided in Table S2. Approval for this study was provided by the Ethical Review and Approval Committee of Jiangmen Central Hospital (Approval number: 2019-002). All patients provided signed informed consent.

Immunohistochemical staining and analysis

HLF expression in paraffin sections was detected by immunohistochemical staining. The experiments were performed by a professional pathological technician. Immunohistochemistry and scoring of HLF expression were performed as previously described (Chen et al., 2020; Zhang et al., 2017; Wu et al., 2020). The staining index (SI) was scored at 20X magnification using an M8 digital microscope (Precipoint, Garching, Germany). Expressions of HLF were classified into low (SI ≤ 4) or high (SI > 4) subgroups.

Identification of CTCs and analysis

For clinical samples, CTC enrichment was performed using the Human Circulating Tumor Cell (hCTC) Subtraction Enrichment Kit (SHE-011; Cytelligen, San Diego, CA, USA) or the Tumor Fisher CTC Detection Kit (Nanopep Biotechnology, Beijing, China) according to the manufacturer’s instructions with six mL of peripheral blood. The cell mixture was obtained using the same protocol as in a previous study, as well as for CTC identification (Liu et al., 2021). Immunofluorescence staining with CD31, CD45, DAPI, and FISH with CEP8 were performed to identify the CTCs. Cells with CD31−/CD45−/CEP8+/DAPI were recognized as single CTC, and cells with CD31−/CD45−/CEP8+/DAPI ≥ 2 were recognized as CTC cluster. Those cells with CD31−/CD45+/CEP8+/DAPI were regarded as white blood cells (WBC) and with CD31+/CD45−/CEP8+/DAPI were regarded as circulating epidermal cell (CEC). High and low quantities of CTCs were classified into low (CTCs ≤ 3) or high (CTCs > 3) subgroups.

Statistical analyses

The mRNA expression levels of HLF from TCGA are presented as means ± standard deviation (SD) and were analyzed with unpaired t-tests and Tukey’s multiple comparisons test. The chi-square test was used to analyze the relationship between HLF expression and EGFR variation, as well as the relationship between the quantity of CTCs and EGFR variation. Survival curves were plotted using the Kaplan–Meier method and compared using the log-rank test. P < 0.05 was considered significant. All experiments were repeated three times. Significant differences were determined using GraphPad Prism 10.

Results

HLF is upregulated in EGFR-mutated LUAD specimens

Genes associated with EGFR mutations in lung cancer samples were analyzed using TCGA. HLF expression was associated with EGFR mutation, with HLF was significantly upregulated in EGFR-mutated samples (Fig. S1A). Among the EGRF-mutation samples, there were 15 cases of E19del., 18 cases of L858R and 33 cases of other mutation types. HLF was significantly upregulated in samples from the E19del. and L858R mutations compared to the wild type. There was a tendency for an increase in the remaining types of mutation samples, but the difference was not statistically significant (Fig. S1B). In total, 343 clinical LUAD samples were collected from our hospital and subjected to Sanger sequencing. There were 139 cases of EGFR mutation, including 70 cases of E19del., 56 cases of L858R and 13 cases of other types of mutation. Further details are presented Table S1. Table S2 provides details of the sequencing results of the 343 LUAD cases. Immunohistochemical staining of HLF was performed on surgical or puncture specimens from the collected cases. HLF was expressed to varying degrees in both surgical and puncture specimens (Fig. 1A). After scoring, the SI of the HLF was distributed from 0 to 12, as shown in Fig. 1B, with SI = 4 as the grouping criterion. After analysis, HLF was found to be upregulated in patients with EGFR mutations compared to the wild type (P = 0.061) (Fig. 1C). Compared with the HLF low group, the proportion of the HLF high group (SI > 4) was significantly higher in the EGFR mutant group (Fig. 1D). HLF also showed no differences between the different EGFR mutant site samples (Fig. 1E), and the proportion of high HLF was significantly increased in the at E19del group. However, the difference was not significant (P = 0.053, Fig. 1F).

Figure 1 HLF is highly expressed in EGFR-mutated LUAD tissues.

(A) Typical pictures of immunohistochemical staining of HLF in EGFRw.t. and different mutant subtypes of lung ADC tissues. (B) The case number of different immunohistochemical staining index of HLF in lung ADC tissues (n = 343). (C) Immunohistochemical staining index of HLF in EGFR mut. lung ADC tissues compared with wild type (w.t., n = 204; EGFR mut., n = 139). Each bar represents the median values ± quartile values. P value was determined by one-way ANOVA test. (D) The ratios of HLF high and low expression in EGFR mut. lung ADC tissues compared with wild type (w.t., n = 204; EGFR mut., n = 139). P value was determined by one-way ANOVA test. (E) Immunohistochemical staining index of HLF in EGFR mut. lung ADC tissues compared with different EGFR mutant subtypes (w.t., n = 204; E19del, n = 70; L858R, n = 56; other, n = 13). Each bar represents the median values ± quartile values. P value was determined by one-way ANOVA test. (F) The ratios of HLF high and low expression in EGFR mut. lung ADC tissues compared with different EGFR mutant subtypes (w.t., n = 204; E19del, n = 70; L858R, n = 56; other, n = 13). P value was determined by one-way ANOVA test.

The number of CTCs was negatively correlated with HLF expression, but not with EGFR mutation

Using a fully automated scanning fluorescence microscope, single-nucleated cells were tagged with CD31, CD45, and CEP8. The phenotypes of CEC, WBC, and CTC were respectively marked with CD31+CD45−CEP8+, CD31−CD45+CEP8+, and CD31−CD45−CEP8+. The results demonstrated that CTCs were effectively separated and collected from peripheral blood (Fig. 2A). The distribution of cases is shown in Fig. 2B. The number of cells was employed as a differentiation metric, with more than three cells being classified as the HLF high group and three or fewer cells being classified as the HLF low group. One to sixteen CTCs were captured from the collected cases, and the number of CTCs was not related to EGFR mutation (Figs. 2C, 2E). In comparison, the percentage of high or low CTCs was also not significantly associated with EGFR mutation, despite gene locus mutation (Figs. 2D, 2F). When HLF high or low was used as a distinction, there was a significant increase in CTCs captured in HLF low cases compared to HLF high (Fig. S2A), and the percentage of CTCs was significantly higher in patients with HLF low expression (Figs. S2B, S2C). These results confirmed that the number of CTCs was not associated with EGFR mutations but was inversely correlated with HLF expression.

Figure 2 The number of CTCs was not associated with EGFR mutations.

(A) Identification of CTCs and CEC in LUAD patients. DAPI, blue; CD45, red; 31CD, yellowish green; CEP8, orange. (B) The case number of CTCs in LUAD patients. (C) CTCs number in EGFR mutant LUAD patients compared with wild type (w.t., n = 204; EGFR mut., n = 139). Each bar represents the median values ± quartile values. P value was determined by one-way ANOVA test. (D) The ratios of CTC high and low in EGFR mutant LUAD patients compared with wild type (w.t., n = 204; EGFR mut., n = 139). P value was determined by one-way ANOVA test. (E) CTCs number in different EGFR mutant subtypes of LUAD patients compared with wild type (w.t., n = 204; E19del., n = 70; L858R, n = 56; other, n = 13). Each bar represents the median values ± quartile values. P value was determined by one-way ANOVA test. (F) The ratios of CTC high and low in different EGFR mutant subtypes of LUAD patients compared with wild type (w.t., n = 204; E19del, n = 70; L858R, n = 56; other, n = 13). P value was determined by one-way ANOVA test.

The expression of HLF and the number of CTCs were significantly associated with the prognosis of LUAD patients

Follow-up of 343 LUAD cases revealed that EGFR mutations had no effect on the degree of pathological differentiation of the patients’ tumor size, metastasis, or stage, as documented in detail in Table S3. In contrast, HLF expression level was closely associated with tumor progression. By comparison, the number of cases with poorly differentiated tumors, large tumor size, tumor metastasis, and advanced stage were statistically higher in patients in the low HLF group; detailed information is presented in Table S4. After Univariate COX regression analysis, the factors associated with progression-free survival in patients with LUAD were tumor stage, tumor size, proximal metastasis, distant metastasis, HLF expression, and the number of CTCs (Fig. 3A). Detailed follow-up information and comparative analyses are presented in Tables S3 and S4. The results of the Multivariate COX regression analysis demonstrated that high HLF expression was an independent protective factor for progression-free survival in patients with lung adenocarcinoma (P < 0.001), whereas high CTCs were an independent risk factor for recurrence or death in patients with LUAD (P < 0.001) (Fig. 3B). Patients with high HLF expression in tumor tissues showed longer survival, with a median survival of 55.2 months (Fig. 3C). In contrast, more CTCs captured in the peripheral blood was associated with a poor prognosis, with a median survival of 28.7 months (Fig. 3D). When low HLF expression and high CTCs coexisted (two points), patients had the shortest median survival of 20.1 months (Fig. 3E). When low HLF or high CTC levels appeared alone (1 point), the median patient survival was 53.3 months. If high HLFs and low CTCs were present together (0 points), the median survival exceeded 60 months. Metastasis is one of the leading causes of death in cancer patients. As ‘seed cells,’ CTCs are closely related to tumor metastasis. Patients with high CTCs had a much higher rate of metastasis than those with low (Fig. 3F). Patients with a high HLF showed the possibility of reduced metastasis, with a lower rate of metastasis than those with a low HLF (Fig. 3G).

Figure 3 COX regression analysis and progression-free survival of all LUAD patients.

(A) Univariate COX regression analysis shows that T, N and M classification, stage, HLF and CTCs are significantly correlated with PFS (n = 343). (B) Multivariate COX regression analysis shows that HLF and CTCs are significantly correlated with PFS (n = 343). P value was determined by log-rank test. HR indicates hazard ratio; 95%CI indicates 95% confidence interval. (C) Kaplan–Meier analysis of progression-free survivals of all LUAD patients with HLF low and high expression (HLF low, blue, n = 177; HLF high, red, n = 166). (D) Kaplan–Meier analysis of progression-free survivals of all LUAD patients with CTCs low and high (CTCs low, blue, n = 213; CTCs high, red, n = 130). (E) Kaplan–Meier analysis of progression-free survivals of all LUAD patients. Patients were divided into 0, 1, and 2 point groups as HLF low and CTCs high taking 1 point singly but 0 point on the contrary (0 point, blue, n = 84; 1 point, red, n = 139; 2 point, green, n = 120). (F) The ratios of metastasis in CTCs high and low of all LUAD patients. P value was determined by one-way ANOVA test. (G) The ratios of metastasis in HLF high and low expression of all LUAD patients. P value was determined by one-way ANOVA test.

In EGFR wild-type cases, univariate COX regression analysis showed that tumor size, distant metastases, HLF expression, and the number of CTCs were correlated to progression-free survival (Fig. 4A). Multivariate COX regression analysis revealed that HLF high expression was an independent protective factor for progression-free survival in patients with LUAD (P = 0.005), whereas CTCs high was an independent risk factor (P = 0.005) (Fig. 4B). The median survival time of patients with high HLF expression was 54.4 months, which was significantly longer than 21.9 months of patients with low HLF group (Fig. 4C). Patients in the CTCs high group had a shorter median survival time of 24.0 months (Fig. 4D). When low HLF expression and high CTCs coexisted (2 points), the patients had the worst median survival, 21.7 months. When low HLF or high CTCs appeared alone (1 point), the median survival was 30.6 months, and if high HLF and low CTCs were present together (0 points), the median survival was greater than 60 months (Fig. 4E). Patients with elevated CTC levels of CTCs experienced a significantly higher incidence of metastasis compared to those with lower CTCs levels (Fig. 4F). Conversely, high HLF expression exhibited the potential for decreased metastasis, with a reduced metastasis rate compared to that of low HLF (Fig. 4G).

Figure 4 COX regression analysis and progression-free survival of EGFR wild type LUAD patients.

(A) Univariate COX regression analysis shows that N classification, HLF and CTCs are significantly correlated with PFS (n = 204). (B) Multivariate COX regression analysis shows that HLF and CTCs are significantly correlated with PFS (n = 204). P value was determined by log-rank test. HR indicates hazard ratio; 95%CI indicates 95% confidence interval. (C) Kaplan–Meier analysis of progression-free survivals of EGFRw.t. LUAD patients with HLF low and high expression (HLF low, blue, n = 109; HLF high, red, n = 95). (D) Kaplan–Meier analysis of progression-free survivals of EGFRw.t. LUAD patients with CTCs low and high (CTCs low, blue, n = 73; CTCs high, red, n = 131). (E) Kaplan–Meier analysis of progression-free survivals of EGFRw.t. LUAD patients. Patients were divided into 0, 1, and 2 point groups as HLF low and CTCs high taking 1 point singly but 0 point on the contrary (0 point, blue, n = 45; 1 point, red, n = 78; 2 point, green, n = 81). (F) The ratios of metastasis in CTCs high and low of EGFRw.t. LUAD patients. P value was determined by one-way ANOVA test. (G) The ratios of metastasis in HLF high and low expression of EGFRw.t. LUAD patients. P value was determined by one-way ANOVA test.

In cases with EGFR mutations, Univariate COX regression analysis revealed that tumor pathologic grade, proximal metastases, HLF expression, and the number of CTCs were correlated with progression-free survival (Fig. 5A). Multivariate COX regression analysis revealed that high HLF expression was an independent protective factor (P < 0.001), whereas high CTC level was an independent risk factor (P = 0.014) (Fig. 5B). Patients exhibiting high HLF expression demonstrated a median survival time of 55.2 months, which was markedly longer than the 28.7 months observed in the low HLF group (Fig. 5C). Conversely, the median survival time for patients in CTCs high group was notably reduced, to 35.4 months (Fig. 5D). When low HLF expression and high CTCs coexisted (2 points), the patients had the shortest median survival, 20.2 months. When HLF low or CTCs high appeared alone (1 point), the median survival was 55.2 months, while HLF high and CTCs low presented together (0 point), the median survival exceeded 60 months (Fig. 5E). Individuals with high CTC counts were more likely to develop metastases than those with low CTC (Fig. 5F). In contrast, patients with high HLF levels showed a lower metastasis rate than those with low HLF group (Fig. 5G).

Figure 5 COX regression analysis and progression-free survival of EGFR mutant ADC patients.

(A) Univariate COX regression analysis shows that N classification, HLF and CTCs are significantly correlated with PFS (n = 139). (B) Multivariate COX regression analysis shows that HLF and CTCs are significantly correlated with PFS (n = 139). P value was determined by log-rank test. HR indicates hazard ratio; 95%CI indicates 95% confidence interval. (C) Kaplan–Meier analysis of progression-free survivals of EGFR-mut. LUAD patients with HLF low and high expression (HLF low, blue, n = 82; HLF high, red, n = 57). (D) Kaplan–Meier analysis of progression-free survivals of EGFR-mut. LUAD patients with CTCs low and high (CTCs low, blue, n = 82; CTCs high, red, n = 57). (E) Kaplan–Meier analysis of progression-free survivals of EGFR-mut. LUAD patients. Patients were divided into 0, 1, and 2 point groups as HLF low and CTCs high taking 1 point singly but 0 point on the contrary (0 point, blue, n = 39; 1 point, red, n = 61; 2 point, green, n = 39). (F) The ratios of metastasis in CTCs high and low of EGFR-mut. LUAD patients. P value was determined by one-way ANOVA test. (G) The ratios of metastasis in HLF high and low expression of EGFR-mut. LUAD patients. P value was determined by one-way ANOVA test.

In summary, the expression of HLF and the number of CTCs are the main factors affecting the PFS and survival time of patients with LUAD, regardless of the presence or absence of EGFR mutations, and they are the main indicators of LUAD prognosis.

Discussion

Overall, our study demonstrated that HLF and CTCs could be prognostic biomarkers for LUAD. They can indicate prognosis regardless of gene mutation and can be used as markers for the precise diagnosis and treatment of LUAD. According to the immunohistochemical scoring criteria we described (Chen et al., 2020), the additional detection of HLF during histopathological diagnosis is feasible, inexpensive, convenient, and does not require additional tissue extraction. The detection of CTCs is becoming increasingly convenient with the use of fully automated capture microscopy, with little financial burden, and is very friendly for oncology patients.

Currently, precision medicine is increasingly used in the prediction, diagnosis, and treatment of cancer owing to the development of biomarkers, tumor immunology, and targeted therapies. Oncogene-driven tumors involving EGFR, ALK, BRAF, and KRAS are highly effective targeted therapeutic agents (Yang et al., 2022). ctDNA, one of the main modalities of liquid biopsy, is widely used for post-treatment monitoring of non-small cell lung cancer and the detection of molecular mechanisms of resistance to TKI treatment, with good sensitivity and specificity (Desai et al., 2024). However, ctDNA assays are affected by various factors, including coagulation, individual DNA enzyme activity, leukocyte DNA interference, blood drawing, and temperature (Lone et al., 2022). More importantly, ctDNA testing is often used for lung cancer prognosis driven by pathogenic gene mutations, as its sensitivity is not sufficiently obvious for patients without gene mutations. In contrast, in the current study, we found that HLF expression and the number of CTCs were independently associated with the prognosis of LUAD regardless of the presence of EGFR oncogenic mutations. This compensates for the lack of tissue and liquid biopsies in patients without oncogenic mutations.

HLF, a member of the proline and acidic amino acid-rich basic leucine zipper transcription 5 factor family (PAR bZIP) (Reszka & Zienolddiny, 2018), plays an important role in neurodevelopment (Hitzler et al., 1999) and fibroblast apoptosis (Suzuki et al., 2018). Furthermore, HLF was proved relevant to cancer, including hematological malignancy (Wahlestedt et al., 2017) and glioma (Chen et al., 2016). In our previous study, HLF expression was dramatically reduced in patients with early recurrence and metastasis of NSCLC, and HLF upregulation inhibited the early recurrence and metastasis of NSCLC (Chen et al., 2020). Bioinformatics research has shown that HLF is associated with NSCLC, and low HLF expression indicates poor prognosis (He & Zuo, 2019). However, there was no further evidence, such as case follow-ups or experimental research, to support these results. In this study, the data of patients diagnosed with LUAD at Jiangmen Central Hospital from 2017 to 2019 were collected for IHC, CTCs capture, and case follow-up, and demonstrated that low HLF-low expression is an independent risk factor for LUAD.

CTCs refer to cancer cells that leave the solid tumor, enter the bloodstream, and contain a range of metastatic precursors, which are essential for cancer progression and are a key step in the spread of cancer cells through circulation to distant non-malignant tissues. CTCs are used as diagnostic biomarkers to identify cancers at an early stage and avoid overtreatment of inert tumors (Castro-Giner & Aceto, 2020; Lawrence et al., 2023). In our study, individual CTCs and CTC clusters were successfully captured, and their numbers were strongly correlated with tumor progression and inversely correlated with HLF expression in tumor tissues. In univariate and multivariate Cox regression analyses, low expression of HLF and high levels of CTCs were independent risk factors affecting the prognosis of patients with LUAD; when both were present, the median survival time of patients was extremely low, at 21.7 months. In conclusion, our study demonstrated that the evidence for the combination of HLF and CTCs as prognostic biomarkers for LUAD is strong and independent of oncogenic mutations.

This study has some limitations. Although we confirmed an inverse correlation between HLF expression and the number of CTCs, we did not illustrate the relevance of HLF expression in CTCs activity or the formation of distant metastatic foci. We will continue our subsequent studies to elucidate the effect of HLF expression in CTC on cell activity and the formation of metastatic foci and to illustrate its role in the prognosis of LUAD. It is well known that the expression of PD-L1 by tumor cells enables them to evade killing by immune cells and promotes tumor progression. Three monoclonal antibodies that target PD-L1 (atezolizumab, durvalumab, and avelumab) are approved for clinical use, including for non-small cell lung cancer (Ramamurthy et al., 2024). One study reported that HLF is one of the key genes affecting the prognosis of advanced head and neck squamous carcinoma with high PD-L1 expression (Dai et al., 2021). Exploring the relationship between HLF and PD-L1 expression in CTCs is extremely important for determining the prognosis of LUAD.

Conclusions

HLF expression and the number of CTCs can be used as effective biomarkers for the prognosis of LUAD with or without pathogenic EGFR carcinogenic mutations. HLF and CTCs can conjointly serve as important guides for the determination, treatment selection, and prognosis of LUAD.

Supplemental Information

Supplemental Information 1 HLF is highly expressed in EGFR-mutated LUAD tissues from TCGA

(A) mRNA expression of HLF in EGFR-mutated LUAD tissues from TCGA (w.t., n = 440; mut., n = 66;). (B) mRNA expression of HLF in different EGFR-mutated subtypes of lung ADC tissues from TCGA (w.t., n = 440; E19del., n = 15; L858R, n = 18; other, n = 33). Each bar represents the median values ± quartile values. P value was determined by one-way ANOVA test.

Supplemental Information 2 The number of CTCs was inversely correlated with HLF expression

(A) CTCs number in HLF high expression LUAD patients compared with HLF low expression (HLF high., n = 177; HLF low, n = 166). Each bar represents the median values ± quartile values. P value was determined by one-way ANOVA test. (B) The ratios of CTCs low and CTCs high in LUAD patients with HLF high or low expression (HLF high, n = 177; HLF low, n = 166). P value was determined by one-way ANOVA test. (C) The ratios of HLF low and high in LUAD patients with CTCs high or low expression (CTCs high, n = 213; CTCs low, n = 130). P value was determined by one-way ANOVA test.

Supplemental Information 3 Supplementary Tables

Supplemental Information 4 Raw data

We would like to thanks Beijing Nanopep Biotechnology Co., Ltd for providing the experimental guidance and Pathology Department of Jiangmen central hospital for experimental helping and instruction. We would like to thank Editage for English language editing too.

Additional Information and Declarations

Competing Interests

Author Contributions

Human Ethics

Data Availability

Xin Zhang is an Academic Editor for PeerJ.

Yaofeng Zhi performed the experiments, analyzed the data, prepared figures and/or tables, authored or reviewed drafts of the article, and approved the final draft.

Jinhua Wu analyzed the data, authored or reviewed drafts of the article, and approved the final draft.

Ronggang Li analyzed the data, authored or reviewed drafts of the article, and approved the final draft.

Xuefei Chang performed the experiments, analyzed the data, prepared figures and/or tables, and approved the final draft.

Silin Liu analyzed the data, prepared figures and/or tables, and approved the final draft.

Wenjie Lu performed the experiments, prepared figures and/or tables, and approved the final draft.

Mingzhu Zheng performed the experiments, prepared figures and/or tables, and approved the final draft.

Baoyi Liu performed the experiments, prepared figures and/or tables, and approved the final draft.

Jiarong Chen conceived and designed the experiments, prepared figures and/or tables, and approved the final draft.

Xin Zhang conceived and designed the experiments, authored or reviewed drafts of the article, and approved the final draft.

Yanming Huang conceived and designed the experiments, authored or reviewed drafts of the article, and approved the final draft.

The following information was supplied relating to ethical approvals (i.e., approving body and any reference numbers):

Institutional Research Ethics Committee of the Jiangmen Central Hospital

The following information was supplied regarding data availability:

The raw data are available in the Supplementary Files.

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
