# Peer review of "A combination of hepatic leukemia factor and circulating tumor cells serve as effective biomarkers for lung adenocarcinoma prognosis"

_PeerJ, doi:10.7717/peerj.19092_

## Round 0.1 · original submission · Major Revisions

The reviewers found the study's results interesting for the field. However, functional and biological explanations of the findings still need to be included. The major and minor concerns of the reviewers should, therefore, be carefully answered.

Reviewer 1 ·

Basic reporting

This study, through bioinformatics analysis and immunohistochemistry, identified the significant value of Hepatic leukemia factor (HLF) in the diagnosis and prognosis assessment of lung adenocarcinoma (LUAD). Additionally, it enhanced the diagnostic efficacy of HLF through the integration of peripheral blood liquid biopsy technology to detect the circulating tumor cells (CTCs). Meanwhile, this research indicated that the expression of HLF and the number of CTCs could be jointly used as effective biomarkers for the prognosis of LUAD with or without the pathogenic EGFR carcinogenic mutation, and HLF and CTCs could conjointly serve as important guides for determination, treatment selection and prognosis in LUAD. Although some of these findings are interesting, there are a number of concerns that need to be addressed.
1. Line 66-67: Grammar error: Unclear subject in the clause leads to confusion in description;
2. Line 71-72: The sentence has inconsistent tense and the usage of "of" is questionable;
3. Line 79-80: The sentence has some grammar and usage issues: The use of "and" to connect two questions lacks clear logical connection in context; "be better predicted" may not be precise enough, consider a clearer expression, such as "How can tumor prognosis be more accurately predicted in cancers without EGFR mutation or other mutation-driven genes?"
4. Line 110: The usage of "by" is questionable;
5. Line 145-146: The sentence contains grammar and word usage errors, leading to potential confusion in reading; Is there literature support for the choice of staining index, and if not, is it appropriate to use “SI=4” as the grouping criterion?
6. Line 149-151: This sentence has a dangling modifier issue;
7. Line 152-153: The sentence lacks clear logical coherence, leading to potential misunderstanding upon reading;
8. Line 156: “CD31-CD45-CEP8+” is not the marker of WBC as described above;
9. Line 158: “The number of 3 cells” should be revised; Is there any basis for choosing “3” as the grouping criteria?
10. Line 237-239: Dangling modifier;
11. In this study, the grouping of HLF and CTCs is incomplete. For instance, it does not explore the prognostic implications of groups such as HLF high + CTCs high and HLF low + CTCs low in lung cancer, which renders the corresponding conclusions of this study unreliable;
12. The authors emphasized the role of HLF and CTCs to be jointly used for the prognosis of LUAD, but did not discuss the related diagnostic efficacy as biomarkers. It is recommended to construct a combined diagnostic and prognostic assessment model.
13. How to avoid the difference in the number of EGFR mutant groups and wild groups?
14. Result1, the specific mechanism of HLF high expression related to which EGFR mutation type needs to be studied in depth.
15. Result2, for the conclusion of "There was an inverse correlation between the amount of CTCs and HLF expression", the results of the picture are not clearly explained.
16. Result3, why not directly divide the collected lung cancer samples into HLF low-expression and high-expression groups (small or more CTC groups) and analyze their survival rate? Isn't this conclusion more convincing?
17. Result3, in addition to survival rate analysis, prognostic analysis also includes relationships with recurrence, metastasis, etc., which should also be included in the analysis.

Experimental design

no comment

Validity of the findings

no comment

Additional comments

It is recommended that the language be polished carefully.

Reviewer 2 ·

Basic reporting

Disruption of the body's internal clock has been linked to higher susceptibility to various cancers like breast, prostate, colorectal, liver, and non-Hodgkin's lymphoma in recent studies. Substantial evidence indicates that working irregular shifts, which disturbs the body's natural rhythms, harms health. A complex self-regulating system of 'clock' genes controls these rhythms in response to regular environmental changes. One of these genes, hepatic leukemia factor (HLF), is a critical transcription factor dependent on the body's clock and significantly regulates various activities. It is a part of the proline- and acidic amino acid-rich basic leucine zipper protein family. It was initially identified in leukemia patients with abnormal E2–HLF fusion gene expression. Subsequently, it was found to be expressed in liver and kidney cells. Additionally, the HLF gene has been found to have an important regulatory role in the development of multiple cancers, including lung, renal, glioma, liver, and breast cancers.
The language is good and clear to understand, sufficient literature cited in text and is found updated.

Experimental design

Given the aforementioned findings, it is imperative for the authors to apprise the readers about the potential role of the HLF gene in mediating immune infiltration, cell apoptosis, epithelial-to-mesenchymal transition (EMT), and cell cycle pathways in the context of lung cancer.
A comprehensive understanding of these mechanisms may pave the way for novel approaches to combating lung cancer progression and drug resistance, offering promise for advanced treatment strategies.
Whether a sample-size calculation was performed?

Validity of the findings

The HLF gene is expressed at lower levels in cancer patients. In that case, they may potentially exhibit resistance to Paclitaxel, Dasatinib, Docetaxel, AZ628, Z-LLNle-CHO, WH-4-023, Bortezomib, Bleomycin, 17-AAG, MLN4924, Vinblastine, YM155, Vinorelbine, CI-1040, BEZ235, Trametinib, RDEA119, selumetinib, and PD-0325901, while being sensitive to CAL-101 and Navitoclax. What could be the primary constraints in proposing the use of HLF induction and the monitoring of CTCs as a potential strategy for employing them as prognostic indicators for lung cancer?

Additional comments

The manuscript "Hepatic leukemia factor and circulating tumor cells combinedly serve as effective biomarkers for lung adenocarcinoma prognosis" is well written and needs minor revision as mentioned, and can be considered.

Reviewer 3 ·

Basic reporting

no comment.

Experimental design

no comment.

Validity of the findings

no comment.

Additional comments

no comment.

Annotated reviews are not available for download in order to protect the identity of reviewers who chose to remain anonymous.

Reviewer 4 ·

Basic reporting

Zhu et al. has aimed to identify a prognostic biomarker for lung adenocarcinoma and has explored the possibility of establishing the hepatic leukemia factor (HLF) and circulating tumor cells as effective biomarkers for early diagnosis of LUADs. The study is interesting and is clinically relevant ,in light of the limited prognostic biomarkers for LUADs. However, the study suffers from lack of functional validation of the effect of HLF in in vitro or in vivo setting. There is a lack of mechanistic understanding of the effects of HIF on metastasis or tumor growth. Additionally, the clinical patient samples needs to be further analyzed and the manuscript suffers from ambiguous writing. The manuscript is replete with missing words and grammatical errors which needs to be corrected and revised expansively. The article needs a major revision and should address the below points:

Experimental design

1. Why hasn't the authors validated any of the findings from patient samples on LUAD cell line models or checked for the HLF expression on EGFR mutated vs wild type cells and the various NSCLC cell lines by using Western Blot/qPCR? Also, how did the authors arrive at the hypothesis that HLF can be a potential biomarker needs to be elucidated in the introduction.
2. The authors should generate HLF knockout cell lines in a LUAD cell line like NCI-H3255 and inject them in a mice model to evaluate the effects of HLF depletion in lung cancer progression. An dox-inducible knockout system in vivo would also help answer whether the HLF knockout affects the lung cancer metastatic progression.
3. Line 31 needs to be replaced as stating that "there is no cure at present" is scientifically unsound.
4. Multiple previous studies have shown correlation between EGFR mutations and prognosis of LUAD patients. Why does the author find no correlation in their analysis? Does the authors have any specific hypothesis for this rather contradictory results?
5. The authors used a cutoff of SI<=4 as HLF low group and SI>4 as HLF high, also CTC>3 were adjudicated as CTC high group. The authors must provide detailed explanation with regards to how these cutoff were decided upon in the methods section. Is there any previous literature which have used the same cutoffs? If so, then must be cited.
6. In light of the evidence provided in Fig. 1 it would be premature to suggest that HLF is upregulated in EGFR-mutated LUAD. The only figure which modestly substantiates this claim is fig 1.D. The other figures doesnt show any significant differences, hence the authors must substantiate their claim further or rephrase the title on Line 129.
7. The authors must provide quantification of HLF expression across all the sample types in Fig. 1 A. Additionally, how did the authors arrive on the cutoff of the stain index of HLF to divide the patients into low and high groups in Fig. 1B.
8. Please provide the specific table where the EGFR mutated samples mentioned in Line 132-133 has been provided. Also, provide detailed description for the readers of what the T, N .M classification refers to in Fig.3,4 and 5, and explain the differences between the different T, N and M classification provided in Supp. Table 3 and 4.
9. Why did the authors use CEP8+ as a staining for WBCs? They are specific Abs which could have been used along with CD45+ for staining WBCs in Fig. 2A.
10. Please provide a statistical comparison analysis for CTC low with HLF high and low , and CTC high with HLF low and high separately for Supp. Fig. 2B.
11. The authors must provide a survival plot for HLF high alone median survival plot for Fig. 3C, 4C and 5C.
12. Did the authors check for the effect of other common mutations in LUADs such as TP53 and KRAS on HIF ? Additionally, the authors should quantify the Fig. 2A to determine the number of CTCs , WBCs and CECs in the samples stained.

Minor Comments:
1. The entire manuscript suffers from grammatical errors and missing words which needs to be fixed. Clear and comprehensive drafting of the manuscript is required to aid the readers.
2. Multiple mistakes such as Line 239 where there is a word missing exists. Additionally, as example, Fig.1A legend should be referred to as "representative images" and not "typical pictures" Rephrase line 79.

Validity of the findings

Please find all the comments above.

---

## Round 0.2 · accepted · Accept

The authors addressed the issues raised by the reviewers.

Reviewer 4 ·

Basic reporting

The authors have addressed the comments adequately.

Experimental design

The authors have addressed the comments adequately.

Validity of the findings

The authors have addressed the comments adequately.